# Effects of Online, Asynchronous Education Modules on Migraine Severity and Elimination Diet Use Among Higher Education Students: An Observational, Pilot Feasibility Study

**DOI:** 10.3390/nu17152432

**Published:** 2025-07-25

**Authors:** Thanh Thanh T. Vo, Amanda K. Jan, Jeffrey Duong, Jenny Sayaseng, Monica Joy, Emily Andrada, Elizabeth Ekpo, Michelle L. Dossett

**Affiliations:** 1School of Medicine, University of California Davis, Sacramento, CA 95817, USA; 2Department of Emergency Medicine, University of California Davis, Sacramento, CA 95817, USA; 3Department of Neurology, University of California Davis, Sacramento, CA 95817, USA; 4Department of Internal Medicine, University of California Davis, Sacramento, CA 95817, USA

**Keywords:** migraine disorders, headache disorders, elimination diets, feasibility studies, nutrition therapy

## Abstract

**Background/Objectives**: Migraine is a debilitating neurologic disorder with diet-related triggers. No studies exist on education on migraine in conjunction with an elimination diet as a non-pharmacologic management approach. **Methods**: Higher education students who self-reported migraine were enrolled in this observational, pilot feasibility study. At baseline, participants completed questionnaires on demographics, migraine disability, and their understanding of migraine and an elimination diet. After one month of self-paced, asynchronous, online modules, participants were reassessed on their understanding of migraine and an elimination diet. Two months later, participants completed follow-up questionnaires on migraine disability, whether they implemented components of the diet, and any barriers they encountered. **Results**: Of 66 students who completed baseline measures, 33 completed the modules and all questionnaires. Of participants who completed the study, 100% found the modules helpful in learning about migraine and an elimination diet; 57.6% incorporated aspects of the elimination diet into their lives. Participants had significant (*p* < 0.001) increases in knowledge both about migraine and an elimination diet. Participants had a potentially clinically significant decrease (14-point MIDAS drop, *p* = 0.10) in migraine symptoms after completing the educational intervention, with a greater decrease among participants who implemented the elimination diet. **Conclusions**: It is feasible to design and implement an education intervention on diet for higher education students, though loss to follow-up was high in this population. The majority of participants who completed the modules adopted aspects of an elimination diet, indicating its feasibility. Further studies with a larger sample size powered to assess the efficacy of this approach are needed.

## 1. Introduction

Migraine is a debilitating neurological disorder affecting roughly 14% of the world’s population [1]. Headache disorders, including migraine and tension-type headache, were the second-most prevalent Level 3 (i.e., specific) disease and the third-leading cause of years lived with disability in the Global Burden of Disease 2021 study [2]. Moreover, there was no improvement in the burden of headache between 2010 and 2021 [2]. Migraine costs approximately USD 9.2 billion per year in direct healthcare expenditures [3]. Debilitating migraine symptoms include sensoriphobia, nausea and vomiting, headache, cognitive disruption, fatigue, and mood changes [4]. Migraine attacks are known to occur in times of stress [5], poor sleep [6], and poor nutrition [7], though with varying scientific understanding of causality and pathophysiology. Notably, the pathophysiology of migraine is complex with both central and peripheral mechanisms [8].

Despite the pervasive and disabling nature of migraine, 44% of people who meet the International Classification of Headache Disorders criteria for migraine do not have a diagnosis of migraine [9]. Because it is often a whole-body disorder with symptoms overlapping other conditions and because of inherent biases in the healthcare system, patients with migraine may not be accurately diagnosed or treated [10,11]. Seeking diagnosis can be especially difficult for people with low migraine- or health literacy [11]. This disparity highlights the need for migraine education to help deliver effective, equitable care.

Moreover, migraine is the leading neurologic cause of disability among young people 5–19 years old and the second leading cause among those 20–59 years of age [12]. Given these demographics, higher education students are at elevated risk of migraine and migraine-related disability. Compounding this risk, students pursuing higher education are frequently sleep-deprived [13], have constant psychological stressors [14], and often lack the time and/or financial resources to take care of their health [15]. For migraine in particular, one study found that college students had an inaccurate and incomplete understanding of migraine management, though students with migraine had significantly more understanding [16]. Conversely, it is reasonable to expect that migraine can negatively affect educational attainment, quality, and performance, thus uniquely impacting quality of life in this population.

Therefore, it is crucial to study the delivery of migraine education to higher education students. Patient education in the area of diet and migraine is especially lacking, in part because evidence on the role of diet in migraine management is variable and without consensus among the majority of studies. There have been a number of research studies focusing on one specific diet with inconsistent results, such as ketogenic, high-folate, low-fat, modified Atkins, and high-omega-3/low-omega 6 diets [17]. Nevertheless, there is no single “migraine diet” used to treat migraine. An elimination diet, which can be used in the treatment of many disorders, excludes food triggers for a specific disease from the patient’s diet. In the case of migraine, there are many known food triggers that may be related to attack onset, such as chocolate [18]. Unfortunately, few studies have tested the impact of an elimination diet avoiding all commonly reported food triggers on migraine frequency and severity.

To address the unmet need for migraine education among higher education students and education on an elimination diet for migraine, the primary objective of this study was to assess the feasibility and tolerability of an asynchronous online course on migraine and an elimination diet that excluded a broad range of potential migraine triggers. The secondary objectives were to preliminarily assess the impact of the diet on migraine symptoms and to understand the barriers that higher education students face in implementing an elimination diet. Understanding these barriers will help to guide future research on education and diet in migraine as well as patient education initiatives to improve the health of individuals with this disabling condition.

## 2. Materials and Methods

### 2.1. Study Cohort

We conducted a three-month, observational, cohort feasibility study of undergraduate, graduate, and professional students at a large, public American university. The study protocol was approved and granted exempt status by the University of California Davis IRB (IRB #1948555, 6 July 2023). Participants were recruited through flyers and emails distributed across undergraduate and graduate campuses, libraries, the student health center, wellness groups, and extracurricular clubs at the university from September 2023 through January 2024. Flyers and emails directed potential participants to a website to learn more about the study and complete a screening questionnaire. Everyone who passed the screening questionnaire was then provided with a letter of information regarding the study and what it involved and could select a box consenting to participate. Additionally, weekly to monthly reminder emails were sent to encourage participants to watch the modules, complete the questionnaires, and consider trying the elimination diet. Inclusion criteria were being a student at the university, history of self-reported migraine, ability to complete online questionnaires and Canvas modules, and age over 18 years. Medication use for migraine was not considered part of the study inclusion or exclusion criteria, as some students may not have been taking medication due to cost or lack of access to medical care. Students who met the criteria were enrolled and completed the study using Qualtrics survey software (Qualtrics, Provo, UT, USA) and an asynchronous education course on Canvas, a web-based learning management system (Canvas, Salt Lake City, UT, USA). Data collection was completed in May 2024.

### 2.2. Study Procedures

Participants were asked to complete questionnaires at three timepoints: on enrollment (baseline); after completing an online, asynchronous course on migraine and an elimination diet over the course of one month (Time 1); and two months after completing the course (Time 2). Study data were collected and managed using Qualtrics. Study participants were sent a link to each questionnaire and asked to include their name, campus email, and phone number on the questionnaire at each timepoint to facilitate participant tracking and data linkage and ensure a single response at each timepoint. Only study personnel involved in recruitment had access to participant names and contact information. At baseline, data were collected on participant demographics, participants’ understanding of migraine and an elimination diet, and their migraine symptoms using the Migraine Disability Assessment Test (MIDAS). Participants were then granted access to two self-paced, asynchronous modules on Canvas. They had one month to complete the modules, after which they completed the Time 1 questionnaires, which collected data on participants’ understanding of migraine and an elimination diet as well as their opinions of the Canvas course. Over the next two months, participants were encouraged to incorporate aspects of an elimination diet for migraine into their life as desired and tolerable. Two months after completing the modules and Time 1 questionnaires, participants were asked to complete the Time 2 questionnaires, which included the MIDAS, questions on the feasibility of incorporating an elimination diet into their life, and an exploration of barriers to doing so. Participants received a USD 20 gift card upon completing the Time 2 questionnaires in appreciation for their time and participation.

### 2.3. Questionnaires

The Migraine Disability Assessment Test (MIDAS) [19] consists of seven questions about the degree of pain of participants’ migraine attacks, ranging from 1 to 10, and the number of days in the last 3 months that participants (1) missed work/school, (2) had their productivity reduced by half or more, (3) did not do household work, (4) had their household work productivity reduced by half or more, (5) missed family and social activities, and (6) had a headache. Migraine scores are used to categorize disability level. A score of 0–5 corresponds to little or no disability; a score of 6–10 corresponds to mild disability; a score of 11–20 corresponds to moderate disability; and a score of 21 or higher corresponds to severe disability.

The Pre-Education and Post-Education Assessments consist of two sections: Knowledge of Migraine and Knowledge of the Elimination Diet. Their purpose was to assess for changes in participants’ understanding of migraine and the elimination diet after completing the modules.

The Knowledge of Migraine section includes ten statements about migraine and migraine management; it asks participants to rate whether the statement is true or false. Question 11 lists 19 symptoms and asks participants to categorize whether they can be associated with migraine or are never associated with migraine. Question 12 asks about the typical duration of a migraine. A complete list of questions and scoring can be found in Appendix A.

The Knowledge of the Elimination Diet section includes 15 true/false questions. A final question lists several foods and asks participants to determine whether each can be a migraine trigger by answering Yes or No. The full questionnaire can be found in Appendix A.

Total scores for both sections are based on the number of correctly answered questions. The Total Knowledge Score ranges from 0 to 53, the Total Migraine Knowledge Score ranges from 0 to 30, and the Total Elimination Diet Knowledge Score ranges from 0 to 23.

The Course Evaluation form includes three questions with a five-point Likert scale response on the usefulness and practicality of the education modules.

The Two-Month Follow-up Survey consists of two sections: (1) Knowledge, Self-Efficacy, and Adherence and (2) Potential Barriers to Following the Elimination Diet. The measure employs a five-point Likert scale for statements such as “Following the elimination diet makes me anxious”. Participants were instructed to skip the section on barriers if they had not implemented the diet. The full questionnaire can be found in Appendix A.

No questionnaires were used to validate diagnosis of migraine. Due to the pilot, feasibility nature of this study, diagnosis was based entirely on participant self-report.

### 2.4. Sample Size and Statistical Analyses

As this was a pilot, feasibility study, sample size was based on feasibility to recruit and complete data collection within a single academic year based on study personnel availability. All data were analyzed as completed by participants. There were no missing data except in cases in which participants chose not to complete an entire questionnaire. Participant characteristics were summarized with percentages for categorical variables (e.g., gender, race/ethnicity, etc.) and means and standard deviations (SD) for continuous variables (e.g., length of migraine diagnosis). Change in knowledge between Baseline and Time 1 follow-up as well as change in MIDAS scores between Baseline and Time 2 follow-up were computed using paired *t*-tests (statistical significance *p* < 0.05). To account for potential changes in migraine symptoms due to some participants initiating a new medication for migraine between the baseline and Time 2 surveys, sensitivity analyses were performed. Results appeared unchanged when omitting patients who started a new medication and thus are presented using the full sample at follow-up. Participant course evaluation and adherence to the elimination diet were summarized using univariate tabulations of ordinal categorical variables. Data were analyzed with Stata version 18.0 (StataCorp LLC, College Station, TX, USA).

### 2.5. Development and Description of the Asynchronous Education Modules

The online educational course consisted of two modules. The first module contained sections describing what migraine is, risk factors for migraine, common triggers of migraine attacks, the role of an elimination diet in migraine, and a 24-hour diet recall activity. The second module contained sections on phases of the elimination diet, a sample weekly recipe guide, food label guidance, the 24-hour diet recall revisited, and bonus tips and resources. The full written content of the course can be found in Appendix A.

The content was adapted from multiple resources; it was reviewed by an integrative medicine physician and a neurologist who specializes in headache. The modules combined written content, videos of the same written content with illustrations, and interactive activities such as matching games. Participants could either watch the videos or read the text at their own pace. Interactive activities were incorporated throughout the course to maintain participant engagement and reinforce the presented concepts. The modules were only available in English.

## 3. Results

A total of 104 individuals were eligible based on screening and provided consent. Of these, only 66 individuals decided to proceed and completed the baseline questionnaires. Only 50% (*n* = 33) of these participants completed the Time 1 and Time 2 questionnaires. The mean age of participants was 24.3 years, with 71.2% identifying as female (Table 1). The mean duration of migraine history was 7.2 years, and at baseline, 50% of participants were receiving care from a healthcare provider for migraine.

At Time 1, after completing the learning modules, there was a statistically significant increase in total knowledge scores, rising from 22.1 to 26.0. Both migraine knowledge (from 16.4 to 18.5) and elimination diet knowledge (from 5.6 to 7.5) improved significantly (*p* < 0.001) compared to pre-course scores (Table 2). Furthermore, at Time 2, two months after the course, 48.5% of participants completely agreed that they were well-informed on how to implement the diet, and 46.9% reported the course improved their dietary habits (Appendix A).

Similarly, at Time 1, all the participants endorsed finding the course helpful, with the majority (60.6%) completely agreeing that the course was beneficial in learning about migraine and the elimination diet, while 39.4% somewhat agreed (Table 3). Additionally, 48.5% of participants completely agreed and 27.3% somewhat agreed that the course was engaging. Furthermore, 36.4% of participants completely agreed and 39.4% somewhat agreed that a 24-hour diet recall activity improved their understanding of the elimination diet.

At Time 2, two months after completing the learning modules, participants’ willingness to try the elimination diet varied. While 57.6% agreed that they had incorporated the diet into their daily routine (48.5% somewhat agreed and 9.1% completely agreed), 27.3% somewhat disagreed, and 12.1% were neutral (Table 4).

To preliminarily assess whether exposure to the learning modules affected migraine frequency or severity, changes in MIDAS scores between Baseline and Time 2 were examined (Table 5). While no statistically significant changes were observed, there was a trend toward a decreased MIDAS score at Time 2 (*p* = 0.10). Notably, participants who somewhat or completely agreed that they had incorporated the elimination diet saw an 18-point reduction in MIDAS scores compared to participants who did not incorporate the elimination diet and only saw an 8-point reduction in MIDAS scores. These results remained unchanged when participants who started new medications for migraine were excluded (Appendix A). There was no significant difference in change in MIDAS scores when comparing participants who initiated new medications to those who did not (Appendix A).

The most common barriers to fully implementing the elimination diet, as reported by participants who tried the diet, included stress or illness (74.1%), social situations (70.3%), school or life responsibilities (62.9%), stopping the diet when symptoms were in control (59.3%), lengthy planning time (59.2%), the restrictive nature of the diet (55.5%), and traveling (51.8%; Appendix A). While 87.1% of participants agreed that they would recommend the diet to others with migraine, only 26.7% completely agreed (36.7% somewhat agreed) that the benefits outweighed the inconveniences (Appendix A).

## 4. Discussion

This study is the first to examine the feasibility and impact of education on migraine and an elimination diet for higher education students as well as the unique barriers this population faces in implementing an elimination diet. We found that it is feasible to design and implement an educational intervention for this population. Our intervention—an asynchronous, online educational course—resulted in statistically significant increases in knowledge about migraine and the elimination diet and uncovered some of the barriers to engaging in an elimination diet for migraine among higher education students. Additionally, it showed possible clinical improvement in migraine symptoms for participants, especially those who tried the elimination diet, which has not yet been demonstrated in the literature.

### 4.1. Feasibility of Online Modules for Migraine Education for Students

The greatest loss of participants (*n* = 38) occurred after completing the screening and consent but before completing baseline questionnaires or gaining access to the modules. This high attrition demonstrates that regardless of whether the educational intervention was useful or interesting for participants, there are reasons that higher education students may not engage in health education in the first place, including competing obligations, perceived burden of participation, or a lack of interest in the topic. Another significant loss of participants (*n* = 33) occurred during the month in which participants had access to the educational modules. Of those who completed the modules and Time 1 questionnaires, 100% found the modules helpful in learning about migraine and the elimination diet, and 75.8% of them found the modules engaging. In this way, the modules were effective, but there was also room for improvement. Likely, participant attrition during the module period reflects both the inherent difficulties in reaching a student population (similar to the pre-module attrition) and the ability of the modules to engage participants (24.2% of study completers did not find the modules engaging).

Part of the success of the educational intervention and perhaps why students found it helpful was because of the objective increase in knowledge that participants gained. This finding stands as proof-of-concept that an online, asynchronous education course can increase patients’ understanding of their disease and potential treatments, which supports the role of digital health solutions in patient education, especially in a student population.

### 4.2. Feasibility of an Elimination Diet for Migraine in Students

Just over half (57.6%) of participants incorporated aspects of the elimination diet into their daily lives, indicating that for the majority of participants, it was feasible to adopt and integrate certain elements of the diet into their routine. This conclusion is further supported by the fact that 87.1% of participants who completed the study stated that they would recommend the elimination diet to other people with migraine.

For the participants who tried the diet, we also examined the barriers students faced to implementing it. The most commonly reported barriers to following the diet were stress or illness, social situations, school or life responsibilities, stopping the diet when symptoms were in control, lengthy planning time, the restrictive nature of the diet, and traveling (Appendix A). Understanding the most common barriers to implementing an elimination diet can help clinicians tailor patient education and migraine management specifically for student populations; it can also guide future trial design to mitigate participant loss to follow-up. One possible curricular modification includes providing testimonials from users of the elimination diet on how they approach their dietary restrictions in social situations or when they are stressed. Another modification could involve connecting participants with more personalized dietary guidance for replacing their trigger foods with similar alternatives. Gaining insight into the common challenges students face can also help to set more realistic expectations for migraine management and symptom improvement in this population.

### 4.3. Effect of the Educational Intervention on Migraine Symptoms

Our study was not powered to analyze the efficacy of a diet-related educational intervention on migraine severity. The change in MIDAS scores among our participants, however, trended downward after the educational intervention. Moreover, participants who tried the elimination diet had a more than two-fold greater decrease in MIDAS scores compared to those who did not implement the diet. Nonetheless, the decreases in MIDAS scores were not statistically significant, likely due to the small number of participants in this pilot feasibility study.

That said, prior research on the MIDAS found that a change of 4.5 points or more on the MIDAS is clinically important for patients with chronic migraine receiving non-pharmacologic treatment [20] and that routine medical management in one study resulted in a decrease in MIDAS score of 14 points [21], which was the average decrease after our intervention. Another study found that MIDAS changes did not fully reflect symptom improvement in a trial of a CGRP receptor monoclonal antibody [22], indicating that the MIDAS changes after our intervention, while not statistically significant, may represent a clinically significant reduction in symptoms. Nonetheless, given the observational nature of this study, we cannot assess causation, and some of the decrease in migraine symptoms observed may be due to natural history of the disease and regression to the mean. Future studies should assess an educational intervention on a larger scale over a longer duration as well as the efficacy of an elimination diet for migraine on symptom reduction directly.

### 4.4. Strengths and Limitations

The strengths of the study include the use of an asynchronous educational intervention, which allowed students to engage with the course at times convenient for their schedules. This is especially important for individuals with migraine, as prolonged screen use can exacerbate headache [23]. For this reason, our course also included audiovisual options for the content, which was more accessible than text-only content. In addition, this study is one of the first to examine elimination of many of the most commonly reported food triggers for migraine.

A significant limitation of the study was the small sample size and narrow study population. Because this study was designed as an observational, pilot feasibility study, we cannot assess causation and did not have the sample size to power analyses on the efficacy of the intervention on migraine symptoms or of the elimination diet on migraine symptoms. Moreover, we chose to study higher education students because of the demonstrated need for migraine education in this population, but doing so limits the generalizability of our study to wider populations and patients of other ages (the mean age of our participants was 24 years). Diagnosis of migraine was based on participant self-report, which may have resulted in some misclassification bias. However, migraine tends to be under- rather than over-diagnosed [9]. Additionally, we experienced a high rate of loss to follow-up but could not survey participants on why they did not want to continue in the study. Future studies should attempt to understand the barriers to beginning or completing an educational intervention on migraine and could also explore other types of educational interventions that may be more engaging, such as in-person courses, hybrid interventions, or asynchronous courses with extended deadlines. Another potential curricular change to increase retention could include scheduling periodic check-ins with multiple participants to create a sense of community and for them to share their experiences with the intervention and the diet. A final limitation was the lack of information on long-term knowledge retention or opinions on the elimination diet beyond two months.

## 5. Conclusions

In conclusion, migraine is a remarkably prevalent and debilitating disease throughout the world. In this study of an online, asynchronous educational course on migraine and an elimination diet, we report feasibility in implementing an educational intervention among higher education students. Students found the modules helpful for learning about migraine. Objectively, their knowledge on both migraine and the elimination diet improved, and the majority of those who completed the modules tried to incorporate the diet into their everyday lives. Participants, especially those who tried the diet, experienced decreases in MIDAS scores that were not statistically significant but may be clinically significant. Next steps include increasing sample size and widening the participant population to analyze efficacy of an educational intervention on symptom frequency and severity. Future educational interventions for the elimination diet could consider including advice from users of the diet on how to navigate the barriers to adhering to it or could trial more personalized dietary advice for participants. Further studies are needed to improve the delivery of education on migraine disease, migraine treatment, and various prevention approaches in order to alleviate the burden of this highly disabling disease.

## Figures and Tables

**Table 1 nutrients-17-02432-t001:** Participant Characteristics.

	Baseline	Time 1	Time 2
	Mean (SD) or *n* (%)	Mean (SD) or *n* (%)	Mean (SD) or *n* (%)
Total participants	66	33	33
Age (years)	24.3 (0.7)	23.5 (0.8)	23.5 (0.8)
Female	47 (71.2)	23 (69.7)	23 (69.7)
Race/ethnicity			
White, Non-Hispanic	29 (43.9)	16 (48.5)	16 (48.5)
Black, Non-Hispanic	1 (1.5)	0 (0.0)	0 (0.0)
Hispanic, Latino, Spanish	16 (24.2)	8 (24.2)	8 (24.2)
Asian, Non-Hispanic	13 (19.7)	5 (15.2)	5 (15.2)
Multiple race/ethnicity	7 (10.6)	4 (12.1)	4 (12.1)
Length of migraine diagnosis (years)	7.2 (1.0)	6.1 (1.4)	6.1 (1.4)
Seeing a provider for migraine	33 (50.0)	14 (42.4)	14 (42.4)

**Table 2 nutrients-17-02432-t002:** Change in Knowledge Scores (*n* = 33).

	Baseline	Time 1	Difference	*p*-Value
	Mean (SD)	Mean (SD)	Mean (SD)	
Total Knowledge Score	22.1 (0.7)	26.0 (0.4)	4.0 (0.8)	<0.001
Migraine Knowledge Score	16.4 (0.5)	18.5 (0.3)	2.1 (0.5)	<0.001
Elimination Diet Knowledge Score	5.6 (0.4)	7.5 (0.2)	1.9 (0.5)	<0.001

Notes. Total Knowledge Score ranges from 0 to 53; Total Migraine Knowledge Score ranges from 0 to 30; Total Elimination Diet Knowledge Score ranges from 0 to 23.

**Table 3 nutrients-17-02432-t003:** Course Evaluation at Time 1 (*n* = 33).

	Completely Disagree	Somewhat Disagree	Neutral	Somewhat Agree	Completely Agree
	*n* (%)	*n* (%)	*n* (%)	*n* (%)	*n* (%)
1. The course was helpful in learning about migraines and the elimination diet.	0 (0.0)	0 (0.0)	0 (0.0)	13 (39.4)	20 (60.6)
2. The course was engaging.	0 (0.0)	1 (3.0)	7 (21.2)	9 (27.3)	16 (48.5)
3. The 24 h diet recall activity helped improve my understanding of the elimination diet.	0 (0.0)	2 (6.1)	6 (18.2)	13 (39.4)	12 (36.4)

**Table 4 nutrients-17-02432-t004:** Adherence at Time 2 (*n* = 33).

	Completely Disagree	Somewhat Disagree	Neutral	Somewhat Agree	Completely Agree
	*n* (%)	*n* (%)	*n* (%)	*n* (%)	*n* (%)
Adherence					
I have incorporated the elimination diet into my everyday life since completing the workshops.	1 (3.0)	9 (27.3)	4 (12.1)	16 (48.5)	3 (9.1)

**Table 5 nutrients-17-02432-t005:** Change in MIDAS Scores.

	Baseline	Time 2	Difference	*p*
	Mean (SD)	Mean (SD)	Mean (SD)	-Value
**Full Sample (*n* = 33)**				
MIDAS Score	42.7 (6.7)	28.5 (3.8)	−14.2 (8.4)	0.10
**High Adherence** **(*n* = 19)**				
MIDAS Score	43.3 (8.3)	25.1 (4.8)	−18.3 (11.0)	0.11
**Low Adherence** **(*n* = 14)**				
MIDAS Score	41.8 (10.9)	33.4 (6.1)	−8.4 (13.3)	0.54

Note. High Adherence = Adherence score 4 (somewhat agree) or 5 (completely agree); Low Adherence = Adherence score 1 (completely disagree), 2 (somewhat disagree), or 3 (neutral).

## Data Availability

The questionnaires and educational content presented in this study are included in the Appendix A. Additional data available by reasonable request to Dr. Dossett.

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
