# Peer review of "Effects of Online, Asynchronous Education Modules on Migraine Severity and Elimination Diet Use Among Higher Education Students: An Observational, Pilot Feasibility Study"

_nutrients, 2025, doi:10.3390/nu17152432_

Round 1
Reviewer 1 Report
Comments and Suggestions for Authors
The authors reported the feasibility of implementing an educational intervention that included an asynchronous course on migraines and an elimination diet among higher education students. The results showed that knowledge of migraine and the elimination diet improved, though MIDAS scores did not change significantly. I suggest some modifications to further improve understanding.
Comments
The reviewer believes it's inappropriate to discuss findings that haven't been presented in the results section. This specifically pertains to lines 217-220. The discussion and conclusions related to these statements (in the "Discussion" and "Conclusion" sections) lack credibility without the corresponding results. Therefore, I strongly recommend clearly presenting these results.
Please describe the process by which the initial 104 individuals who underwent screening were narrowed down to the 66 who completed the baseline questionnaire.
Please specify how the participants' headaches were diagnosed as migraine.
Line 215,
If you state "greater decrease," please present the results of the statistical analysis between the groups to support this claim.
Line 226 and Table S2
Please clarify the exact number of participants who actually attempted the dietary intervention. Additionally, in Table S2, please explain the discrepancy in the number of respondents for "Social-Emotional" (30 participants) and "Cost and Convenience" (29 participants).
Author Response
We thank the reviewer for taking the time to read our manuscript and provide thoughtful comments. Our responses to the comments are below. The line numbers refer to the newly uploaded tracked changes Microsoft Word document.
Response to Reviewer 1
Comment 1: "The reviewer believes it's inappropriate to discuss findings that haven't been presented in the results section. This specifically pertains to lines 217-220. The discussion and conclusions related to these statements (in the "Discussion" and "Conclusion" sections) lack credibility without the corresponding results. Therefore, I strongly recommend clearly presenting these results."
Response 1: Thank you for the suggestion. We have added these results as a new Supplementary Table 2 (text is now lines 238-241, Table S2 is page 44 of the Supplementary Materials).
Comment 2: "Please describe the process by which the initial 104 individuals who underwent screening were narrowed down to the 66 who completed the baseline questionnaire."
Response 2: All 104 individuals who were eligible based on screening were eligible to participate in the study. Unfortunately, only 66 individuals decided to proceed and completed the baseline questionnaires. The study team did not purposefully eliminate anyone between the screening and questionnaire stages and we do not know why these individuals chose not to participate. We have modified the wording on line 198 to clarify this. We comment on this loss of potential participants beginning on line 266 of the Discussion section.
Comment 3: "Please specify how the participants' headaches were diagnosed as migraine."
Response 3: Thank you for asking. Given the pilot, feasibility nature of this study and limited funding, diagnosis of migraine was based on participant self-report as explained in the study inclusion criteria (line 102). While this does pose a risk of misclassification bias (line 340 of the limitations section), we feel this risk is small and migraine tends to be under rather than over diagnosed.
Comment 4: "Line 215, If you state "greater decrease," please present the results of the statistical analysis between the groups to support this claim."
Response 4: Thank you for catching this. That particular statistical analysis was not performed. We tried to minimize multiple testing given the pilot nature of this work. We have modified the wording in lines 236-238 to better reflect the data that is presented.
Comment 5: "Line 226 and Table S2 -- Please clarify the exact number of participants who actually attempted the dietary intervention. Additionally, in Table S2, please explain the discrepancy in the number of respondents for "Social-Emotional" (30 participants) and "Cost and Convenience" (29 participants)."
Response 5: Thank you for asking for clarification. As reported in the manuscript, 33 participants completed the Time 2 questionnaires. Per Supplementary B, participants were instructed to skip the questions on barriers if they did not try the elimination diet. Thus, only 30 participants answered the social-emotional questions and only 29 completed the questions on cost and convenience. We have added a footnote to the supplementary table (now Table S3, page 45 of Supplementary Materials) for clarification.

Reviewer 2 Report
Comments and Suggestions for Authors
Dear authors,
Thank you for the opportunity to revise your observational study “Effects of Online, Asynchronous Education Modules on Migraine Severity and Elimination Diet Use Among Higher Education Students: An Observational, Pilot Feasibility Study”.
Please find some specific suggestion to improve your article below.
Abstract
- I suggest to expand the results section with more numeric value, moreover I suggest to check if all your keywords are MeSh terms
Introduction
- migraine is characterized by a complex physiopathology that is part of their burden. In fact, the pathophysiology of migraine involves both central and peripheral mechanisms, i.e. peripheral and central sensitization, lack of habituation, thalamo-cortical dysrethmia, hyperexcitability of the motor cortex. The burden of migraine and their physiopathology should be expanded in order to support your work. I suggest one reference that could support this paragraph, concerning generally pathophysiology of migraine (doi: 10.1152/physrev.00034.2015.) and the second one more recent concerning neurophysiological value of sensitization and habituation in migraine (doi: 10.1097/WNP.0000000000001055.)
- Please delineate more precisely the specific research gap that the study aims to fill
- I would split your aim in a primary and secondary aims
Method
- I would specify the timeline when the study was conducted
- Were the Checklist for Reporting Results of Internet E-Surveys (CHERRIES) guidelines respected? Please specify
- was the STROBE check list adopted?
- were the diagnostic criteria of Headache Disorders-3 (ICHD-3) used by an expert neurologist?
- please clarify in inclusion/exclusion criteria if subject take prophylactic or symptomatic medication, this is an important variable
-The manuscript does not clearly justify the chosen sample size or discuss how it ensures sufficient power to detect expected results
Results
The quality of tables is very good, but in a different font respect to the main text. I suggest to use the same font
Discussion
- I find this section too long, especially the fist paragraph is a repetition of the introduction. I suggest to revise this section: first paragraph describe begetter your main findings; second paragraph comparing your result with previous studies; third add a more comprehensive discussion of the study's limitations; fourth more thorough consideration of the study's generalizability would help readers assess the findings' applicability; finally, it could be interesting suggesting future work.
- Please avoid the use of p-value in this section, it represents a repletion of the results section. I suggest to use p-value and other numeric value only in the results section.
Author Response
We thank the reviewer for taking the time to read our manuscript and provide thoughtful comments. Our responses to the comments are below. The line numbers refer to the newly uploaded tracked changes Microsoft Word document.
Response to Reviewer 2:
Comment 1: “Abstract - I suggest to expand the results section with more numeric value, moreover I suggest to check if all your keywords are MeSh terms.”
Response: We thank the reviewer for the suggestion to expand the results section with more numeric values. Unfortunately, we are already at the word limit for the abstract and cannot add additional information. We also appreciate the excellent suggestion to use MeSh terms for keywords, and yes, all of our keywords are MeSh terms.
Comment 2: “Introduction - migraine is characterized by a complex physiopathology that is part of their burden. In fact, the pathophysiology of migraine involves both central and peripheral mechanisms, i.e. peripheral and central sensitization, lack of habituation, thalamo-cortical dysrethmia, hyperexcitability of the motor cortex. The burden of migraine and their physiopathology should be expanded in order to support your work. I suggest one reference that could support this paragraph, concerning generally pathophysiology of migraine (doi: 10.1152/physrev.00034.2015.) and the second one more recent concerning neurophysiological value of sensitization and habituation in migraine (doi: 10.1097/WNP.0000000000001055.)
Response 2: We appreciate the reviewer’s scholarly suggestions and have added a sentence with the first reference to the introduction as requested (lines 47-48; reference 8). We have decided not to discuss the pathophysiology of migraine in depth, or the clinical outcomes in episodic migraine outlined in the second reference, as these are really peripheral to the discussion of a dietary intervention that was not intended to target or specifically examine migraine pathophysiology.
Comment 3: Introduction - Please delineate more precisely the specific research gap that the study aims to fill. I would split your aim in a primary and secondary aims.”
Response 3: Thank you for the suggestion. We have modified the wording in the last paragraph of the introduction to better clarify our research objectives (lines 80-85).
Comment 4: “Method - I would specify the timeline when the study was conducted”
Response 4: Thank you for the suggestion. We have added this information in lines 95-96 of the Methods.
Comment 5: “Method - Were the Checklist for Reporting Results of Internet E-Surveys (CHERRIES) guidelines respected? Please specify. Was the STROBE check list adopted?”
Response 5: Yes, the STROBE checklist was followed. Thank you for making us aware of the CHERRIES guidelines. We have incorporated the relevant information from CHERRIES into our revision.
Comment 6: “Method - were the diagnostic criteria of Headache Disorders-3 (ICHD-3) used by an expert neurologist?”
Response 6: Given the pilot, feasibility nature of this study and limited funding, diagnosis of migraine was based on participant self-report as explained in the study inclusion criteria (line 102). While this does pose a risk of misclassification bias (limitations section, line 340), we feel this risk is small and migraine tends to be under rather than over diagnosed.
Comment 7: “Method - please clarify in inclusion/exclusion criteria if subject take prophylactic or symptomatic medication, this is an important variable”
Response 7: Thank you for asking. Medication use was not considered part of the study inclusion or exclusion criteria as some students may not be taking medication regularly due to cost or lack of access to medical care. We have added this information to lines 103-105.
Comment 8: “Method -The manuscript does not clearly justify the chosen sample size or discuss how it ensures sufficient power to detect expected results”
Comment 8: Thank you for catching this unintended omission. Sample size was not powered for statistical significance given the pilot, feasibility nature of this study. Rather, it was based on feasibility to recruit and complete data collection within a single academic year as the primary authors were medical students at the time. We have added sample size information to lines 166-169.
Comment 9: “Results - The quality of tables is very good, but in a different font respect to the main text. I suggest to use the same font”
Response 9: Thank you for pointing out this difference. The text by default in the journal template is a serif font. We feel that tables are much easier to read in sans serif fonts. We will defer to the journal regarding their standard font preferences.
Comment 10: “Discussion - I find this section too long, especially the fist paragraph is a repetition of the introduction. I suggest to revise this section: first paragraph describe begetter your main findings; second paragraph comparing your result with previous studies; third add a more comprehensive discussion of the study's limitations; fourth more thorough consideration of the study's generalizability would help readers assess the findings' applicability; finally, it could be interesting suggesting future work.
Response 10: Thank you for the suggestions. We have eliminated the first paragraph and condensed other portions of the discussion. The first paragraph now summarizes the main results. We feel the more detailed discussion regarding feasibility of the online modules and the diet as well as the effect on MIDAS scores are important. The paragraph on study strengths is now more clear and we have added more information about limitations. Future work is suggested in both the limitations and conclusions paragraphs.
Comment 11: “Discussion - Please avoid the use of p-value in this section, it represents a repletion of the results section. I suggest to use p-value and other numeric value only in the results section.
Response 11: Thank you for the suggestion. We have eliminated all p-values in the discussion section.

Round 2
Reviewer 1 Report
Comments and Suggestions for Authors
The authors have diligently addressed the reviewers' comments and revised the manuscript clearly. Please consider one additional improvement.
My previous Comment 3 may have been unclear. While I understand that information of headaches was collected via a questionnaire, I suggest clarifying how migraine diagnoses were subsequently established. Specifically, did you diagnose migraine based on the questionnaire data according to ICHD-3 criteria?
Author Response
Reviewer 1 Comment: “My previous Comment 3 may have been unclear. While I understand that information of headaches was collected via a questionnaire, I suggest clarifying how migraine diagnoses were subsequently established. Specifically, did you diagnose migraine based on the questionnaire data according to ICHD-3 criteria?”
Response: No questionnaires were used to validate diagnosis of migraine. Due to the pilot, feasibility nature of this study, diagnosis was based entirely on participant self-report. We have added this information to the revised manuscript (lines 167-168).

Reviewer 2 Report
Comments and Suggestions for Authors
well done
